# AN EFFICIENT ALGORITHM FOR ENTROPIC OPTIMAL TRANSPORT UNDER MARTINGALE-TYPE CONSTRAINTS

## ABSTRACT

This work introduces novel computational methods for entropic optimal transport (OT) problems under martingale-type conditions. The problems can map to a prevalent class of OT problems with structural constraints, encompassing the discrete martingale optimal transport (MOT) problem, as the (super-)martingale conditions are equivalent to row-wise (in-)equality constraints on the coupling matrix. Inspired by the recent empirical success of Sinkhorn-type algorithms, we propose an entropic formulation for the MOT problem and introduce Sinkhorn-type algorithms with sparse Newton iterations that utilize the (approximate) sparsity of the Hessian matrix of the dual objective. As exact martingale conditions are typically infeasible, we adopt entropic regularization to find an approximate constraint satisfied solution. We show that, in practice, the proposed algorithms enjoy both super-exponential convergence and robustness with controllable thresholds for total constraint violations.

## 1 INTRODUCTION

Obtaining the martingale optimal transport (MOT) (Peyré et al., 2019) plan between statistical distributions has attracted significant research interests (Tan & Touzi, 2013; Beiglböck et al., 2013; Galichon et al., 2014; Dolinsky & Soner, 2014; Guo & Obłój, 2019). In the quantized setting, one encodes the martingale condition into two matrices $V, W \in \mathbb{R}^{n \times d}$, and the MOT problem admits a linear programming (LP) formulation:

$$\min_{P: P\mathbf{1}=\mathbf{r}, P^\top \mathbf{1}=\mathbf{c}, P \geq 0} C \cdot P,$$
$$\text{subject to } PV = W, \tag{1}$$

where $\cdot$ stands for entry-wise inner product, $C \in \mathbb{R}^{n \times n}$ is the cost matrix, and $\mathbf{r} = [r_1, \ldots, r_n]^\top, \mathbf{c} = [c_1, \ldots, c_n]^\top \in \mathbb{R}^n$ are respectively the source and target density with $\sum_i r_i = \sum_j c_j = 1$. Likewise, super-martingale conditions (Nutz & Stebegg, 2018) in optimal transport can be written as an LP

$$\min_{P: P\mathbf{1}=\mathbf{r}, P^\top \mathbf{1}=\mathbf{c}, P \geq 0} C \cdot P,$$
$$\text{subject to } PV \geq W. \tag{2}$$

The martingale-type conditions $PV = W$ and $PV \geq W$ constitute a prevalent class of constrained optimal transport problems whereby a few equality or inequality constraints are placed for every site in the source distribution. The large number of constraints makes the corresponding optimization task quite different from optimal transport (OT). There has been a considerable body of work on the mathematical property of MOT (Ghoussoub et al., 2019; Huesmann & Trevisan, 2019; Alfonsi et al., 2020; Backhoff-Veraguas & Pammer, 2022; Wiesel, 2023) and OT problems with super martingale conditions (Nutz & Stebegg, 2018). The MOT problem under entropic regularization is of independent research interest and is studied as a Schrodinger bridge problem with martingale constraints (Henry-Labordere, 2019; Nutz & Wiesel, 2024).

**Motivation** Super-martingale conditions are prevalent for optimal transport problems with inequality constraints from inventory management concerns (Galichon, 2018). In this case, the source distribution and target distribution, respectively, model receiver and supplier in a transport model. For example, suppose that the resource supplied by supplier $j$ is of an auxiliary utility $v_j$ and the receiver $i$ needs the total utility to exceed $w_i$. In this case, write $\mathbf{v} = [v_1, \ldots, v_n]^\top, \mathbf{w} = [w_1, \ldots, w_n]^\top$ and the structural constraint on the coupling matrix $P$ reads

$$P\mathbf{v} \geq \mathbf{w}.$$

The MOT problem appears first in financial applications in computing upper and lower bounds for model-free option pricing under the calibrated market model. This task assumes an asset with known initial and final distributions and an exotic option whose expected payoff is a function of the price of the asset at the initial and final time. In the calibrated market model, the distribution of the asset is a martingale, which gives rise to the martingale condition in MOT. The computed bounds are model-free as they hold under all stochastic processes the asset undergoes.

In addition, fairness in machine learning is growing in importance as a benchmark for machine learning algorithms (Mehrabi et al., 2021; Barocas et al., 2023). Fairness in resource allocation can also be cast as martingale-type constraints similar to Si et al. (2021); Buyl & De Bie (2022).

**Main approach** Inspired by recent theoretical analysis and empirical success of the Sinkhorn's algorithm (Yule, 1912; Sinkhorn, 1964; Cuturi, 2013) for optimal transport, this work uses entropic regularization and explores fast numerical algorithms for the OT problems with martingale-type constraints in equation 1 and equation 2. In contrast to recent works in constrained optimal transport with entropic regularization (Benamou et al., 2015; Tang et al., 2024a), the constraints considered in this setting are special with its $nd$ constraints embedded in the linear equation $PV = W$ or $PV \geq W$.

For OT under martingale-type constraints, entropic regularization following Fang (1992) admits a dual formulation in the form of a concave maximization problem. Given the formulation, we develop two iterative maximization algorithms with a per-iteration complexity of $O(n^2)$. We introduce a Sinkhorn-type algorithm utilizing the sparse structure of the Hessian matrix. We further utilize the fact that the full Hessian matrix admits sparse approximations and introduce a Sinkhorn-Newton-Sparse (SNS) algorithm, which performs Sinkhorn-type iterations followed by sparse Newton iterations. The SNS algorithm rapidly converges to the entropically optimal solution, in practice often achieving exponential or even super-exponential convergence. Thus, the numerical performance of the proposed approach has the same $O(n^2)$ per-iteration complexity, and we show it has similar practical convergence properties as that of Sinkhorn's algorithm in optimal transport.

For equality constraints of the type $PV = W$, it often occurs that the LP does not admit a feasible solution. Thus, we propose to consider a modified LP problem for MOT, first introduced in Guo & Obłój (2019), with the defining feature that it allows for the transport plan to have constraint violations under a threshold. For practical purposes, having control over the constraint violation threshold has the additional benefit that it allows more flexibility in the obtained transport plan.

**Contribution** We summarize our contribution as follows:

- For the MOT problem, we propose a novel entropic regularization approach based on approximate constraint satisfaction.
- Following the analysis in Weed (2018), we prove that the entropically optimal MOT solution is exponentially close to the LP solution.
- We show that the approximate Hessian sparsity in Tang et al. (2024a) extends to the martingale-type constraint setting.
- We introduce a Sinkhorn-type algorithm and a Sinkhorn-Newton-Sparse algorithm for OT under martingale constraints and super-martingale constraints.

## 1.1 RELATED LITERATURE

**Model-free option pricing** There is a large body of work on martingale optimal transport in option pricing. The readers are referred to Tan & Touzi (2013); Beiglböck et al. (2013); Galichon et al.

(2014); Dolinsky & Soner (2014); Guo & Obłój (2019) for detailed derivations. In general, an option might depend on multiple assets and more than two time steps, which would necessitate a multi-marginal martingale optimal transport (MMOT) framework, as can be seen in Eckstein et al. (2021); Nutz et al. (2020). We remark that multi-marginal OT is exponentially hard to compute even under entropic regularization (Lin et al., 2022), and the same is true for MMOT. Thus, even though our framework readily applies to the MMOT case by taking entropic LP regularization, we shall not pursue this direction in this work.

**Constrained optimal transport** Constrained optimal transport (Peyré et al., 2019; Tang et al., 2024a) describes optimal transport tasks under equality or inequality constraints, with MOT and partial optimal transport (Chapel et al., 2020; Le et al., 2022; Nguyen et al., 2022; 2024) being two of the most widely considered cases. In particular, iterative Bregman projection is a widely used methodology for solving OT problems with equality and inequality constraint (Benamou et al., 2015). This work is more similar to Tang et al. (2024a), which uses a variational framework derived directly from the entropic LP formulation. The main contribution of this work compared to Tang et al. (2024a) is that this work applies to a setting in which one has $O(n)$ equality constraints. In contrast, Tang et al. (2024a) has an explicit assumption that only $O(1)$ constraints are allowed for efficiency consideration. Moreover, this work applies to a setting in which one controls the total constraint violation, which is quite different from the formulation of the aforementioned works, which are all based on exact constraint satisfaction.

**Variational methods in optimal transport** There is considerable research interest in the variational form of entropic OT (Dvurechensky et al., 2018; Lin et al., 2019; Kemertas et al., 2023; Tang et al., 2024b). Similar to the OT case, this work provides a variational framework that converts entropic MOT to a convex optimization problem, for which a wide range of existing tools can be used. Detailed analysis of the dual potential of the entropic MOT problem could lead to convergence bounds of the methods provided.

## 1.2 NOTATIONS

For $n \in \mathbb{N}$, we let $[n] = \{1, \ldots, n\}$. We use $M \cdot M' := \sum_{ij} m_{ij} m'_{ij}$ to denote the entry-wise inner product. For a matrix $M$, the notation $\log(M)$ stands for entry-wise logarithm, and similarly $\exp(M)$ denotes entry-wise exponential. We use the symbol $\|M\|_1$ to denote the entry-wise $l_1$ norm, i.e. $\|M\|_1 := \|\text{vec}(M)\| = \sum_{ij} |m_{ij}|$. The $\|M\|_\infty$ and $\|M\|_2$ norms are defined likewise as the entry-wise $l_\infty$ and $l_2$ norms, respectively. The notation $\mathbf{1}$ denotes the all-one vector of appropriate size.

## 2 BACKGROUND

### 2.1 MARTINGALE-TYPE CONDITIONS

Constraint of the type $PV = W$ or $PV \geq W$ often arises from continuous geometric problems. From continuous distributions $\mu, \nu \in L^2(\mathbb{R}^d)$, the discretization step approximates the two distributions by weighted samples. For $\mu, \nu$, one typically performs sampling or quantization to obtain points $\{\mathbf{w}_i\}_{i=1}^n, \{\mathbf{v}_j\}_{j=1}^n \subset \mathbb{R}^d$ with weights $\{r_i\}_{i=1}^n, \{c_j\}_{j=1}^n$. The discretization is through point mass approximation by taking $\mu \approx \hat{\mu} = \sum_{i=1}^n r_i \delta_{\mathbf{w}_i}, \nu \approx \hat{\nu} = \sum_{j=1}^n c_j \delta_{\mathbf{v}_j}$.

After the discretization, the MOT problem becomes a discrete optimization task with the decision space being coupling matrices $P \in \mathbb{R}_{\geq 0}^{n \times n}$. For a coupling matrix $P$, we use $(X, Y) \sim P$ to denote that $(X, Y)$ is a pair of random variables with $\mathbb{P}[X = \mathbf{w}_i, Y = \mathbf{v}_j] = p_{ij}$. We require the marginal distribution of $X, Y$ to equal $\hat{\mu}, \hat{\nu}$, which coincides with the row/column sum condition $P\mathbf{1} = \mathbf{r}, P^\top \mathbf{1} = \mathbf{c}$ in optimal transport. The defining feature of the MOT problem is that one requires the joint distribution $(X, Y)$ to be a martingale, i.e., $\mathbb{E}_{(X,Y) \sim P}[Y \mid X = \mathbf{w}_i] = \mathbf{w}_i$, which one can write in terms of the coupling matrix $P$ by $\frac{1}{\sum_j P_{ij}} \sum_j P_{ij} \mathbf{v}_j = \mathbf{w}_i$. We use the condition that $\sum_j P_{ij} = r_i$, and so the discretized martingale condition is

$$\sum_j P_{ij} \mathbf{v}_j = r_i \mathbf{w}_i. \tag{3}$$

By taking $V = [\mathbf{v}_1, \ldots, \mathbf{v}_n]$ and $W = [r_1 \mathbf{w}_1, \ldots, r_n \mathbf{w}_n]^\top$, we see that the martingale condition in equation 3 is equivalent to $PV = W$. Likewise, the super-martingale condition is modelled by $\mathbb{E}_{(X,Y) \sim P}[Y \mid X = \mathbf{w}_i] \geq \mathbf{w}_i$, which can be written as the condition $PV \geq W$.

## 2.2 SUPER-MARTINGALE CONDITIONS IN E-COMMERCE RANKING

The multi-objective ranking is a formulation where the goal is to find a ranking of products that perform well in multiple relevance metrics (Dong et al., 2010; Dai et al., 2011; Momma et al., 2019; Carmel et al., 2020). The task is practically relevant and is an important instance of information retrieval (Liu et al., 2009; Manning, 2009). In practice, the objectives might be conflicting with no ranking satisfying all of the objectives. To this end, one applies a convex relaxation which can be interpreted as a stochastic ranking policy of the form $(p_l, \pi_l)_{l \in [L]}$ where $\sum_{l \in [L]} p_l = 1$ and the policy picks ranking $\pi_l$ with probability $p_l$. The use of optimal transport arises in this context by considering the doubly stochastic matrix $P = \sum_{l \in [L]} p_l \pi_l$. The entry $p_{ij}$ is the probability of assigning product $i$ to position $j$.

For linear additive ranking metrics, such as precision, recall, and discounted cumulative gain (DCG), the expectation of the performance of the ranking policy only depends on $P$. Thus, optimization over such linear metrics in expectation is an optimal transport task, and constrained optimal transport instances occur when one places certain linear metrics as equality or inequality constraints. Once $P$ is solved, one can recover a stochastic policy $(p_l, \pi_l)_{l \in [L]}$ through the Birkhoff algorithm with $L = O(n^2)$ (Birkhoff, 1946).

In addition, even though ranking to a user is primarily deterministic in practice, the optimal stochastic ranking can provide useful information for ranking design. For example, one might consider the expected position of each product $j$ in the optimal stochastic ranking, and the quantity is computable through the equation $\sum_k P_{kj} k$. We remark that this average position calculation is the barycentric projection under the transport $P$ (Villani et al., 2009).

For the stochastic ranking policy, the super-martingale condition usually occurs as diversity constraints. For example, when the user searches for a type of product, it might make sense to present products that are complementary to the searched product type (McAuley et al., 2015). Moreover, the product ranking case has inherent product heterogeneity in the sense that the complementary products might be of varying degrees of relevance to the searched product. The product information is encoded by a vector $\mathbf{v} = [v_1, \ldots, v_n]^\top$, where $v_i$ encodes the extent to which product $i$ belongs to the complementary product type. Thus, the subgroup diversity requirement is modeled by a constraint

$$P\mathbf{v} \geq \mathbf{w},$$

where $\mathbf{w} = [w_1, \ldots, w_n]^\top$ encodes the threshold at each position $i \in [n]$.

*Remark* 1. The mathematical structure of the martingale and super-martingale conditions are similar. For simplicity, subsequent sections in the main text focus on MOT. The super-martingale condition is a simpler case and is deferred to Appendix D.

## 3 APPROXIMATE CONSTRAINT SATISFACTION IN MOT

While it might be conceptually appealing to enforce the $O(n)$ equality constraints in equation 1, this formulation faces significant feasibility and robustness concerns. For feasibility, there is a simple observation supporting the claim: Summing over the martingale condition in equation 3 for all $i \in [n]$ shows that $\hat{\mu}, \hat{\nu}$ must coincide in their respective barycenter, i.e., $\sum_i r_i \mathbf{w}_i = \sum_j c_j \mathbf{v}_j$, which is quite strict and one can see any perturbation might change a feasible LP scheme into an infeasible one. The criteria for feasibility is more complicated than coinciding barycenter, and practical post-processing of the discretization to maintain feasibility is an open problem in general. In addition, exact constraint satisfaction in MOT faces robustness concerns even when the problem is feasible. The construction in Brückerhoff & Juillet (2022) shows that MOT problems with exact constraint satisfaction are unstable: when $d \geq 2$, the optimal cost from equation 1 might fail to converge even when $(\hat{\mu}, \hat{\nu}) \to (\mu, \nu)$. Thus, entropic LP algorithms based on equation 1 are prone to feasibility and stability issues coming from discretization errors in general.

For the martingale condition, this work focuses on an approximate constraint satisfaction approach due to the reasons discussed. For a threshold parameter $\varepsilon > 0$, we write the program as follows:

$$\min_{P:P\mathbf{1}=\mathbf{r},P^\top\mathbf{1}=\mathbf{c},P\geq 0} C \cdot P,$$
$$\text{subject to } \|PV - W\|_1 \leq \varepsilon, \tag{4}$$

where $\|M\|_1$ denotes the entry-wise $l_1$ norm, i.e. $\|M\|_1 := \|\text{vec}(M)\| = \sum_{ij}|m_{ij}|$. Moreover, we write equation 4 in an equivalent LP formulation with an auxiliary variable $E \in \mathbb{R}^{n \times d}$:

$$\min_{\substack{P,E:P\mathbf{1}=\mathbf{r},P^\top\mathbf{1}=\mathbf{c},P,E\geq 0,\mathbf{1}^\top E\mathbf{1}\leq\varepsilon \\ PV-W-E\leq 0,PV-W+E\geq 0}} C \cdot P. \tag{5}$$

We remark that the choice of $\varepsilon$ has an overall simple rule from Guo & Obłój (2019). Let $W_1$ be the Wasserstein-1 distance based on the $l_1$ metric in $\mathbb{R}^d$. Let $\delta = W_1(\mu, \hat{\mu}) + W_1(\nu, \hat{\nu})$, and then any choice of $\varepsilon \geq \delta$ leads to a feasible LP problem. Moreover, when the cost matrix $C$ comes from a cost function $h\colon \mathbb{R}^d \times \mathbb{R}^d \to \mathbb{R}$ with $c_{ij} = h(\mathbf{w}_i, \mathbf{v}_j)$, the estimation of continuous martingale optimal transport cost through equation 5 has an estimation error of $\text{Lip}(h)\varepsilon = O(\varepsilon)$, where $\text{Lip}(h)$ is the Lipschitz constant of the cost function $h$. Hence, the approximate constraint satisfaction construction is robust and feasible under a suitable choice of $\varepsilon$.

*Remark* 2. In practice, $\mu, \nu$ are continuous distributions with compact support, and $\hat{\mu}, \hat{\nu}$ are obtained through quantization or sampling. If $\hat{\mu}, \hat{\nu}$ are obtained through quantization, the bound on $\delta$ can be obtained through conventional error analysis in histogram-based density estimation (Wasserman, 2006). If $\hat{\mu}, \hat{\nu}$ are obtained through sampling, one would need to determine $\varepsilon$ through cross-validation, and alternatively one can use upper bounds for $\delta = W_1(\mu, \hat{\mu}) + W_1(\nu, \hat{\nu})$ such as in Weed & Bach (2019); Chewi et al. (2024).

**Entropic formulation**    We use the entropic LP formulation in Fang (1992) to add entropy regularization. In particular, one writes

$$\min_{\substack{P,S,T,E,q:P\mathbf{1}=\mathbf{r},P^\top\mathbf{1}=\mathbf{c} \\ S=W-PV+E \\ T=PV-W+E \\ \mathbf{1}^\top E\mathbf{1}+q=\varepsilon}} C \cdot P + \frac{1}{\eta} H(P,S,T,E,q), \tag{6}$$

where $S, T \in \mathbb{R}^{n \times d}, q \in \mathbb{R}$ are auxiliary slack variables, and the entropy term is defined by

$$H(P,S,T,E,q) = \sum_{ij} p_{ij}\log(p_{ij}) + \sum_{i\in[n],k\in[d]} e_{ik}\log(e_{ik}) + s_{ik}\log(s_{ik}) + t_{ik}\log(t_{ik}) + q\log(q).$$

We refer to equation 6 as the entropic MOT problem. In particular, the entropic LP approach leads to an exponential convergence guarantee by Theorem 1, which shows that the entropy-regularized optimal solution is exponentially close to the optimal solution (proof is in Appendix B):

**Theorem 1.** *For simplicity, assume that $\sum_i r_i = \sum_j c_j = 1$ and that the LP in equation 5 has a unique solution $P^\star$. Denote $P_\eta^\star$ as the entropically optimal transport plan in equation 6. There exists a constant $\Delta$, depending only on the LP in equation 5, so that the following holds for $\eta \geq \frac{1+3\varepsilon(1+\log(3nd+1))}{\Delta}$:*

$$\|P_\eta^\star - P^\star\|_1 \leq 6n^2(1+3\varepsilon)\exp\left(\frac{-\eta\Delta + 3\varepsilon\log(3nd+1)}{1+3\varepsilon}\right).$$

We remark that typically $\varepsilon \ll 1$, and so the exponential convergence guarantee is quite close to that of entropic optimal transport (Weed, 2018). In addition, one may use different entropic regularization strengths for the terms in equation 6, which might lead to practical performance benefits.

## 4 MAIN ALGORITHM

### 4.1 VARIATIONAL FORMULATION OF ENTROPIC MOT

By introducing Lagrangian dual variables and using the minimax theorem (derivation is standard and deferred to Appendix C), one obtains the associated dual problem to equation 6:

$$
\begin{aligned}
\max_{\mathbf{x},\mathbf{y},A,B,u} f(\mathbf{x},\mathbf{y},A,B,u) = & -\frac{1}{\eta} \sum_{ij} \exp\left(\eta(-c_{ij} + \sum_{k\in[d]} (a_{ik}+b_{ik})v_{jk} + x_i + y_j) - 1\right) \\
& + \sum_i x_i r_i + \sum_j y_j c_j + \sum_{i\in[n],k\in[d]} (a_{ik}+b_{ik})w_{ik} + \varepsilon u - \frac{1}{\eta}\exp(\eta u - 1) \\
& - \frac{1}{\eta}\left[\sum_{i\in[n],k\in[d]} \exp(\eta a_{ik}-1) + \exp(-\eta b_{ik}-1) + \exp(\eta(u-a_{ik}+b_{ik})-1)\right],
\end{aligned}
\tag{7}
$$

where the optimization over $f$ is an unconstrained maximization task with variables $\mathbf{x} \in \mathbb{R}^n, \mathbf{y} \in \mathbb{R}^n, A \in \mathbb{R}^{n\times d}, B \in \mathbb{R}^{n\times d}, u \in \mathbb{R}$. Intuitively, the $x,y$ variables correspond to the row and the column constraint, and the $A, B, u$ variables correspond to the approximate satisfaction of the martingale condition. As a consequence of the minimax theorem, maximizing over $f$ is equivalent to solving the problem defined in equation 6. We emphasize that $f$ is *concave*, and thus, one can use routine convex optimization techniques to solve the problem.

Among the alternative implementations, a notable candidate is the adaptive primal-dual accelerated gradient descent (APDAGD) algorithm, which has shown robust performance in optimal transport (Dvurechensky et al., 2018). We leave the detail to Appendix A for APDAGD on entropic MOT. Overall, our proposed Sinkhorn-type algorithms enjoy better performance for converging to entropically optimal solutions.

### 4.2 SINKHORN-TYPE ALGORITHM

We introduce the implementation of the Sinkhorn-type algorithm for entropic MOT. Similar to Sinkhorn's algorithm for entropic optimal transport, we let $\mathbf{g} = (\mathbf{x}, A, B, u)$ and split the dual variables into $\mathbf{y}$ and $\mathbf{g}$. The Sinkhorn-type algorithm performs an alternating maximization on $(\mathbf{y}, \mathbf{g})$. The algorithm is summarized in Algorithm 1. The optimization in $\mathbf{y}$ has an explicit solution by the formula on Line 7 of Algorithm 1. For the optimization on $\mathbf{g}$, we show later in this section that $\nabla_{\mathbf{g}}^2 f$ has $O(n)$ nonzero entries. Thus, for the $\mathbf{g}$ variable, the maximization over $\mathbf{g}$ uses Newton's method with back-tracking line search (Boyd & Vandenberghe, 2004). The Newton step iteration count takes $N_{\mathbf{g}} = 1$ for simplicity, and we observe the iteration count is sufficient for good numerical performance.

**Sparsity of Hessian** We show that $\nabla_{\mathbf{g}}^2 f$ has only $O(n)$ nonzero entries. We write $A = [\mathbf{a}_1, \ldots, \mathbf{a}_d]$ and $B = [\mathbf{b}_1, \ldots, \mathbf{b}_d]$, and let $P = \exp\left(\eta(-C + (A+B)V^\top + \mathbf{x}\mathbf{1}^\top + \mathbf{1}\mathbf{y}^\top) - 1\right)$ denote the intermediate transport plan obtained from the current dual variable. Direct calculation shows

$$
\nabla_{\mathbf{x}}\nabla_{\mathbf{b}_k} f = \nabla_{\mathbf{x}}\nabla_{\mathbf{a}_k} f = -\eta \operatorname{diag}(P\mathbf{v}_k),
$$

and one can likewise show that the blocks $\nabla_{\mathbf{a}_k}\nabla_{\mathbf{b}_k} f, \nabla_{\mathbf{a}_k}\nabla_{\mathbf{a}_{k'}} f, \nabla_{\mathbf{b}_k}\nabla_{\mathbf{b}_{k'}} f$ are diagonal matrices. Essentially, the sparsity structure arises from the fact that in $f$ the dual variables $x_i, a_{ik}, b_{ik}$ are only non-linearly coupled with dual variables $x_i$ and $\{b_{ik'}, a_{ik'}\}_{k'\in[K]}$. Lastly, the block $\nabla_{\mathbf{g}}\nabla_u f$ only introduces $O(n)$ non-zero entries.

**Complexity analysis of Algorithm 1** We omit the scaling in $d$ for simplicity, as typically $d = O(1)$. The $\mathbf{y}$ update step is the well-understood column scaling step in Sinkhorn's algorithm with an $O(n^2)$ complexity. For the $\mathbf{g}$ update step, as $\nabla_{\mathbf{g}}^2 f$ has only $O(n)$ nonzero entries, one can apply a sparse linear solver to obtain $\Delta\mathbf{g}$ in $O(n^2)$ time. Moreover, querying $f$ and $\nabla f$ are both $O(n^2)$ operations. In summary, the per-iteration complexity is $O(n^2)$ as that of Sinkhorn's algorithm.

*Remark* 3. It is also possible to split the dual variable into $\mathbf{x}, \mathbf{y}, \mathbf{h} = (A, B, u)$ and perform alternating maximization on $(\mathbf{x}, \mathbf{y}, \mathbf{h})$. Similarly, the update for the $\mathbf{x}, \mathbf{y}$ variable can be performed by

---

**Algorithm 1** Sinkhorn-type algorithm for entropic MOT

---

**Require:** $f, \mathbf{x}_{\text{init}} \in \mathbb{R}^n, \mathbf{y}_{\text{init}} \in \mathbb{R}^n, A_{\text{init}}, B_{\text{init}} \in \mathbb{R}^{n \times d}, u_{\text{init}} \in \mathbb{R}, N, i = 0, N_{\mathbf{g}} = 3$
1: $\mathbf{y} \leftarrow \mathbf{y}_{\text{init}}, \mathbf{g} \leftarrow (\mathbf{x}_{\text{init}}, A_{\text{init}}, B_{\text{init}}, u_{\text{init}})$             ▷ Initialize dual variable
2: **while** $i < N$ **do**
3:      $i_g \leftarrow 0, i \leftarrow i + 1$
4:      # Column scaling step
5:      $(\mathbf{x}, A, B, u) \leftarrow \mathbf{g}$
6:      $P = \exp\left(\eta(-C + (A + B)V^\top + \mathbf{x}\mathbf{1}^\top + \mathbf{1}\mathbf{y}^\top) - 1\right)$
7:      $\mathbf{y} \leftarrow \mathbf{y} + \left(\log(c) - \log(P^\top \mathbf{1})\right)/\eta$
8:      # $\mathbf{g}$ variable update step
9:      **while** $i_g < N_{\mathbf{g}}$ **do**
10:          $\Delta \mathbf{g} = -\left(\nabla^2_{\mathbf{g}} f\right)^{-1} \nabla_{\mathbf{g}} f$             ▷ Obtain search direction
11:          $\alpha \leftarrow \text{Line\_search}(f, \mathbf{g}, \Delta \mathbf{g})$
12:          $\mathbf{g} \leftarrow \mathbf{g} + \alpha \Delta \mathbf{g}$
13:          $i_g \leftarrow i_g + 1$
14:      **end while**
15: **end while**
16: Output dual variables $(\mathbf{x}, \mathbf{y}, A, B, u)$.

---

matrix scaling, and the update for $\mathbf{h}$ can be done by Newton's method. Overall, we observe better numerical performance for alternating maximization on $(\mathbf{y}, \mathbf{g})$.

### 4.3 SPARSE NEWTON ALGORITHM

For enhanced accuracy, we augment Algorithm 1 with an efficient Newton's method which optimizes over dual variables jointly. Sparse Newton iteration performs Newton's method with the Hessian matrix replaced by its sparsification, which is a type of quasi-Newton method (Nocedal & Wright, 1999; Tang et al., 2024b).

The sparse Newton iteration is motivated by an approximate sparsity analysis of the Hessian matrix of dual potential, which shows that the Hessian matrix admits accurate sparse approximation. We define important concepts for subsequent approximate sparsity analysis. Let $\|\cdot\|_0$ denote the $l_0$ norm. The *sparsity* of a matrix $M \in \mathbb{R}^{m \times n}$ is defined by $\tau(M) := \frac{\|M\|_0}{mn}$. Furthermore, we say that a matrix $M \in \mathbb{R}^{m \times n}$ is $(\lambda, \delta)$-sparse if there exists a matrix $\tilde{M}$ so that $\tau(\tilde{M}) \leq \lambda$ and $\|M - \tilde{M}\|_1 \leq \delta$.

**Approximate sparsity of Hessian**    Let $P$ be the intermediate transport plan formed by the current dual variable. We show that the approximate sparsity of the Hessian matrix $\nabla^2 f$ reduces to that of $P$. By previous discussion, the blocks $\nabla^2_{\mathbf{y}} f, \nabla^2_{\mathbf{g}} f$ are sparse for $\mathbf{g} = (\mathbf{x}, A, B, u)$. Thus, we only focus on the block $\nabla_{\mathbf{y}} \nabla_{\mathbf{g}} f$. The block $\nabla_{\mathbf{y}} \nabla_u f$ is negligible as it contributes only $n$ non-zero entries. For the blocks $\nabla_{\mathbf{y}} \nabla_{\mathbf{x}} f, \nabla_{\mathbf{y}} \nabla_{AB} f$, we compute

$$\nabla_{\mathbf{y}} \nabla_{\mathbf{x}} f = -\eta P^\top, \quad \nabla_{\mathbf{y}} \nabla_{\mathbf{b}_k} f = \nabla_{\mathbf{y}} \nabla_{\mathbf{a}_k} f = -\eta \operatorname{diag}(\mathbf{v}_k) P^\top,$$

which shows that one can obtain sparse approximation of $\nabla^2 f$ by sparse approximation of $P$.

We now discuss the approximate sparsity of the transport plan $P$. To reach the super-exponential convergence stage, both Newton's method and quasi-Newton methods rely on the current dual variable to be close to the maximizer. Therefore, the sparse Newton iteration is only used when $P \approx P^\star_\eta$, where $P^\star_\eta$ is the entropically optimal transport plan in equation 6. Therefore, the applicability of sparse Newton iteration to aid super-exponential convergence relies on the approximate sparsity of $P^\star_\eta$. By the fundamental theorem of linear programming, a unique solution to equation 5 must be a basic solution (Luenberger et al., 1984). Then, assuming uniqueness, the optimal coupling matrix $P^\star$ for equation 4 can have $2n - 1 + nd$ nonzero entries. Thus under Theorem 1, $P^\star_\eta$ is $O(\lambda, \delta)$-sparse, where $\lambda = O(1/n)$ and $\delta$ is exponentially small in $\eta$.

In summary, following the Hessian matrix computation and the sparsity pattern of $P^\star_\eta$, one only needs to keep an $O(1/n)$ fraction of entries in the Hessian matrix for an accurate Hessian approx-

imation. The approximate sparsity argument relies on $P \approx P_\eta^\star$. Thus, for practical purposes, it is desirable to perform warm initialization with Algorithm 1 before applying the sparse Newton iterations. In addition to the importance of warm initialization to Newton's method, another important factor particular to this setting is that initialization leads to a better Hessian approximation.

**Algorithm implementation** We introduce Algorithm 2, which is the main algorithm for entropic MOT under the approximate constraint satisfaction formulation. One runs the Sinkhorn-type algorithm in Algorithm 1 for a few iterations, followed by sparse Newton iterations. Following Tang et al. (2024b), we refer to Algorithm 2 as the Sinkhorn-Newton-Sparse algorithm for entropic MOT.

**Hessian approximation details** We specify the sparsification operation $\text{Sparsify}(\nabla^2 f, \rho)$ in Algorithm 2. We first keep the first $\lceil \rho n^2 \rceil$ largest terms in the transport matrix $P$ and obtain the resulting sparsification $P_{\text{sparse}}$. By the discussion given, we only need to perform the sparsification procedure for the blocks $\nabla_\mathbf{y} \nabla_{AB} f, \nabla_\mathbf{y} \nabla_\mathbf{x} f$ and their respective transpose, which is done by replacing the role of $P$ with that of $P_{\text{sparse}}$. In particular, the approximation $\text{Sparsify}(\nabla^2 f, \rho)$ replaces the $\nabla_\mathbf{y} \nabla_\mathbf{x} f$ block with $-\eta P_{\text{sparse}}^\top$ and replaces the $\nabla_\mathbf{y} \nabla_{\mathbf{b}_k} f, \nabla_\mathbf{y} \nabla_{\mathbf{a}_k} f$ blocks with $-\eta \, \text{diag}(\mathbf{v}_k) P_{\text{sparse}}^\top$.

---

**Algorithm 2** Sinkhorn-Newton-Sparse for entropic MOT

---

**Require:** $f, \mathbf{x}_{\text{init}} \in \mathbb{R}^n, \mathbf{y}_{\text{init}} \in \mathbb{R}^n, A_{\text{init}}, B_{\text{init}} \in \mathbb{R}^{n \times d}, u_{\text{init}} \in \mathbb{R}, N_1, N_2, \rho, i = 0$
  1: # Warm initialization stage
  2: Run Algorithm 1 for $N_1$ iterations to obtain warm initialization $(\mathbf{x}_{\text{init}}, \mathbf{y}_{\text{init}}, A_{\text{init}}, B_{\text{init}}, u_{\text{init}})$.
  3: $z \leftarrow (\mathbf{x}_{\text{init}}, \mathbf{y}_{\text{init}}, A_{\text{init}}, B_{\text{init}}, u_{\text{init}})$          ▷ Initialize dual variable
  4: # Newton stage
  5: **while** $i < N_2$ **do**
  6:  $H \leftarrow \text{Sparsify}(\nabla^2 f, \rho)$          ▷ Sparse approximation of $\nabla^2 f$.
  7:  $\Delta \mathbf{z} \leftarrow -H^{-1}(\nabla f(\mathbf{z}))$          ▷ Solve sparse linear system
  8:  $\alpha \leftarrow \text{Line\_search}(f, \mathbf{z}, \Delta \mathbf{z})$          ▷ Line search for step size
  9:  $\mathbf{z} \leftarrow \mathbf{z} + \alpha \Delta \mathbf{z}$
 10:  $i \leftarrow i + 1$
 11: **end while**
 12: Output dual variables $(\mathbf{x}, \mathbf{y}, A, B, u) \leftarrow \mathbf{z}$.

---

**Complexity analysis of Algorithm 2** For the formula of the Hessian, the cost of obtaining and sparsifying the Hessian is $O(n^2)$. By keeping an $O(1/n)$ fraction of entries of the Hessian through sparsification, obtaining the search direction in a sparse linear system solving step has cost $O(n^2)$. Thus, the sparse Newton iteration has a per-iteration complexity of $O(n^2)$.

## 5 Numerical experiment

We conduct three numerical experiments to showcase the performance of the proposed algorithms for entropically regularized optimal transport under martingale and super-martingale conditions. The goal is to obtain an accurate approximation of the LP solution efficiently. Therefore, we take $n = 800$ and $\eta = 1200$. The choice of parameter leads to an interesting setting where the problem size is relatively large and the entropically optimal transport plan is close to the LP solution in transport cost. For the evaluation metric, we form the intermediate transport plan $P$ with the dual variable and compute the $l_1$ distance $\|P - P_\eta^\star\|$. The reference entropically optimal transport plan $P_\eta^\star$ is obtained by running full Newton iteration until convergence. As a benchmark, we also include the performance of the APDAGD algorithm. The detail for the Sinkhorn-type algorithm and the Sinkhorn-Newton-Sparse algorithm for the super-martingale case is in Appendix D.

Similar to interior point methods, directly optimizing under a large $\eta$ without warm initialization is less efficient. Therefore, the experiments use a warm initialization strategy with a geometric scheduling of $\eta$. We take an initial regularization strength of $\eta_0 = 12.5$ and take $N_\eta = \lceil \log_2(\eta/\eta_0) \rceil$. Then, we use successively doubling regularization levels $\eta_0 < \ldots < \eta_{N_\eta}$ so that $\eta_l = 2\eta_{l-1}$ for $l = 1, \ldots, N_\eta - 1$. We run 5 iterations of Algorithm 1 for every regularization level $\eta_l$ for

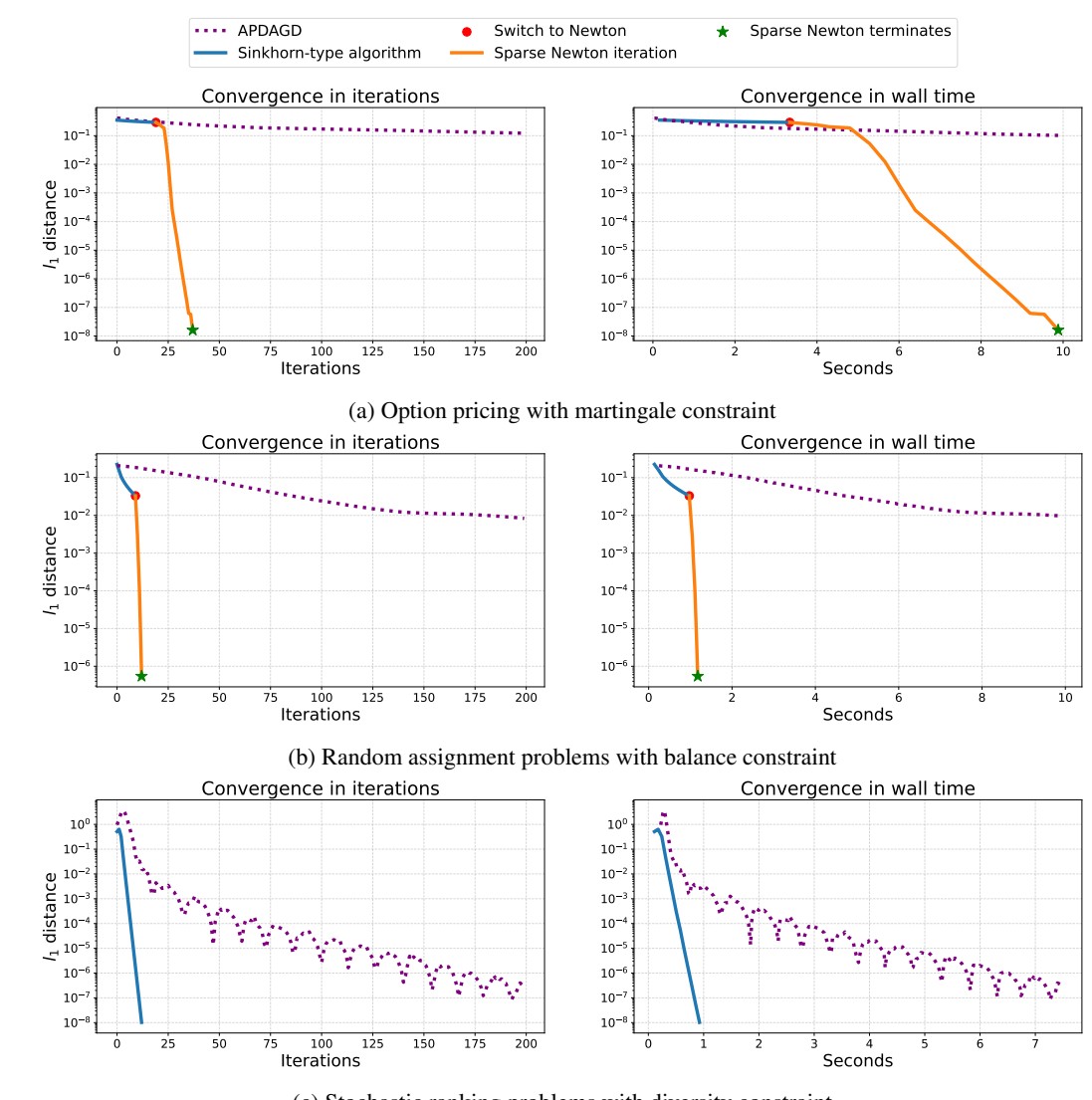

(a) Option pricing with martingale constraint

(b) Random assignment problems with balance constraint

(c) Stochastic ranking problems with diversity constraint

Figure 1: Performance of Algorithm 2 on optimal transport problems with martingale-type constraints. Here, the system size is $n = 800$. A warm initialization is applied for the two MOT examples.

$l = 1, \ldots, N_\eta - 1$. Our warm initialization strategy takes a few iterations and leads to a better optimization landscape at the chosen value of $\eta$. The run time for the warm initialization is quick, and so we omit it in the plotted results for simplicity. To ensure fairness of comparison, the APDAGD algorithm uses the same initialization.

**Option pricing under martingale condition** The first experiment concerns the setting of option pricing under the martingale condition (Hobson & Neuberger, 2012; Guo & Obłój, 2019). In this case, one typically has access to the probability distribution of one asset, which is why we take $d = 1$, even though the algorithm can handle more general cases. We take $\mu$ to be the uniform distribution $\mathrm{Unif}([0,1])$ with $d\mu(x) = \mathbf{1}_{x \in [0,1]}$, and we take $\nu$ to be the law of $X + Y$, where $X \sim \mu, Y \sim \mathcal{N}(0, 10^{-4})$. The source and target distribution are obtained from quantization, and we check that taking $\varepsilon = 2/n = 0.0025$ is sufficient to guarantee feasibility. For the cost we take $C_{ij} = |\mathbf{v}_i - \mathbf{w}_j|$, which is the payoff function considered in Hobson & Neuberger (2012). We plot the result in Figure 1a. We take $N_1 = 20$ and run a few steps of sparse Newton iteration. One can

see that the SNS algorithm is able to achieve convergence to machine accuracy in a few iterations, far exceeding the performance of APDAGD.

**Resource allocation under balance constraints**   We consider a random assignment problem (Aldous, 2001) where we take $\mathbf{r} = \mathbf{c} = \frac{1}{n}\mathbf{1}$ and we generate the entries of the cost matrix $C$ by i.i.d. random variables following the distribution $\mathrm{Unif}([0, 1])$. In resource allocation tasks where one allocates products to customers, one might encounter a *balance constraint* on the transport plan, in which two subgroups of products need to have equal weights sent to each customer. Let $S_A, S_B$ denote the two subgroups, and such a balance constraint can be written by

$$P(\frac{n}{|S_A|}\mathbf{1}_{S_A} - \frac{n}{|S_B|}\mathbf{1}_{S_B}) = 0,$$

where $\mathbf{1}_S$ for $S \subset [n]$ is a one-hot encoding with $(\mathbf{1}_S)_i = 1$ if $i \in S$ and $(\mathbf{1}_S)_i = 0$ otherwise. The constraint is a martingale condition where each index $i$ is an embedding of $v_i = n/|S_A|$ if $i \in S_A$ and $v_i = -n/|S_B|$ if $i \in S_B$.

In our case, as the matrix $C$ does not have a special structure, we simply take $S_A = \{1, \ldots, 100\}$ and $S_B = \{101, \ldots, 200\}$. Also, it is clear that the problem is always feasible, and we take $\varepsilon = 0.1$ to allow for some constraint violation. The result is plotted in Figure 1b. We take $N_1 = 10$ for the number of Sinkhorn-type iterations, and we see that the proposed method has better performance than APDAGD and converges quickly to machine accuracy. As the matrix $C$ is randomly generated, we test the performance across the random instances by repeating the same experiment 100 times. For all of the instances, the SNS algorithm reaches machine accuracy within $N_2 = 5$ sparse Newton iterations.

**Stochastic ranking under diversity constraint**   We consider a stochastic ranking problem with a diversity constraint under the e-commerce setting as described in Section 2.2. In this setting, each product with index $j$ has a main relevance score $s_j$ and an auxiliary utility $v_j$. We consider a normalized discounted cumulative gain metric (NDCG) with $C_{ij} = \alpha \frac{-s_j}{\log_2(1+i)}$, where $\alpha$ is the normalization constant (Järvelin & Kekäläinen, 2002). The diversity constraint for the stochastic ranking policy asks the expected auxiliary utility for each position $i$ exceeds $w_i$. We let $s_j, v_j \sim \mathrm{Unif}([0, 1])$. For the information retrieval setting, the primary focus of the ranking task is on top positions. Therefore, we take the threshold at position $i$ to be $w_i = 0.3$ when $i < 40$, and $w_i = 0$ otherwise. For this case, no warm initialization is applied. The result is plotted in Figure 1c, and we see that $N_1 = 11$ iterations of the Sinkhorn-type algorithm suffices to reach machine accuracy.

## 6   CONCLUSION

We introduce two numerical algorithms for entropic regularization of optimal transport problems under martingale-type constraints. While the two algorithms' numerical performance is quite strong, future work should analyze the proposed approach's convergence property.

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

# A  ACCELERATED FIRST-ORDER METHOD FOR ENTROPIC MOT

In this section, we implement the adaptive primal-dual accelerated gradient descent (APDAGD) method in Dvurechensky et al. (2018) for entropic MOT. In particular, the APDAGD algorithm already generalizes to this case, as it is a general-purpose algorithm for entropic LP. The term $f$ is the dual potential defined in equation 7. The algorithm is summarized in Algorithm 3. The per-iteration complexity of this algorithm is $O(n^2)$. One can also consider alternatives such as in Lin et al. (2019).

---

**Algorithm 3** Adaptive primal-dual accelerated gradient descent algorithm (APDAGD)

---

**Require:** $f, N, k = 0, \mathbf{z}_0 = \boldsymbol{\zeta}_0 = \boldsymbol{\lambda}_0 = 0_{2n+m}$
1: $\alpha_0 \leftarrow 0, \beta_0 \leftarrow 0, L_0 = 1,$
2: **while** $k < N$ **do**
3:     $M_k = L_k/2$
4:     **while** True **do**
5:         $M_k = 2M_k$
6:         $\alpha_{k+1} = \frac{1+\sqrt{1+4M_k\beta_k}}{2M_k}$
7:         $\beta_{k+1} = \beta_k + \alpha_{k+1}$
8:         $\tau_k = \frac{\alpha_{k+1}}{\beta_{k+1}}$
9:         $\boldsymbol{\lambda}_{k+1} \leftarrow \tau_k\boldsymbol{\zeta}_k + (1 - \tau_k)\mathbf{z}_k$
10:         $\boldsymbol{\zeta}_{k+1} \leftarrow \boldsymbol{\zeta}_k + \alpha_{k+1}\nabla f(\boldsymbol{\lambda}_{k+1})$
11:         $\mathbf{z}_{k+1} \leftarrow \tau_k\boldsymbol{\zeta}_{k+1} + (1 - \tau_k)\mathbf{z}_k$
12:         **if** $f(\mathbf{z}_{k+1}) \geq f(\boldsymbol{\lambda}_{k+1}) + \langle \nabla f(\boldsymbol{\lambda}_{k+1}), \mathbf{z}_{k+1} - \boldsymbol{\lambda}_{k+1}\rangle - \frac{M_k}{2}\|\mathbf{z}_{k+1} - \boldsymbol{\lambda}_{k+1}\|_2^2$ **then**
13:             Break
14:         **end if**
15:     **end while**
16:     $L_{k+1} \leftarrow M_k/2, k \leftarrow k + 1$
17: **end while**
18: Output dual variables $(\mathbf{x}, \mathbf{y}, A, B, u) \leftarrow \mathbf{z}_{N-1}$.

---

# B  PROOF OF THEOREM 1

**Definition 1.** Define $\mathcal{P}$ as the polyhedron formed by the feasible set of equation 4, i.e.

$$\mathcal{P} := \{P \mid P\mathbf{1} = \mathbf{r}, P^\top\mathbf{1} = \mathbf{c}, P \geq 0, \|PV - W\|_1 \leq \varepsilon\}.$$

The symbol $\mathcal{V}$ denotes the set of vertices of $\mathcal{P}$. The symbol $\mathcal{O}$ stands for the set of optimal vertex solutions, i.e.

$$\mathcal{O} := \underset{P \in \mathcal{V}}{\arg\min}\, C \cdot P. \tag{8}$$

The symbol $\Delta$ denotes the vertex optimality gap

$$\Delta = \min_{Q \in \mathcal{V} - \mathcal{O}} Q \cdot C - \min_{P \in \mathcal{O}} P \cdot C.$$

We can now finish the proof.

*Proof.* This convergence result is mainly due to the application of Corollary 9 in Weed (2018) to this case. We define another polyhedron $\mathcal{Q}$ as follows:

$$\mathcal{Q} := \{(P, S, T, E, q) \mid P\mathbf{1} = \mathbf{r}, P^\top\mathbf{1} = \mathbf{c}, \mathbf{1}^\top E\mathbf{1} + q = \varepsilon,$$
$$S = W - PV + E \geq 0, T = PV - W + E \geq 0, E \geq 0, q \geq 0\}$$

We use $R_1$ and $R_H$ to denote the $l_1$ and entropic radius of $\mathcal{Q}$ in the sense defined in Weed (2018). We first bound $R_1$. For any $(P, S, T, E, q) \in \mathcal{Q}$ we note that $S, T \geq 0$ and so $\|S\|_1 + \|T\|_1 = \|S + T\|_1 = 2\|E\|_1$. Thus for $R_1$ one has

$$1 \leq R_1 = \max_{(P,S,T,E) \in \mathcal{Q}} (\|P\|_1 + 3\|E\|_1 + |q|) \leq 1 + 3\varepsilon.$$

For $R_H$, we first bound the entropic radius by the entropic radius of $P$ and of $(S, T, E)$.

$$R_H = \max_{(P,S,T,E,q),(P',S',T',E',q')\in\mathcal{Q}} H(P,S,T,E,q) - H(P',S',T',E',q')$$

$$\leq \max_{(P,S,T,E,q),(P',S',T',E',q')\in\mathcal{Q}} H(P) - H(P')$$

$$+ \max_{(P,S,T,E,q),(P',S',T',E',o')\in\mathcal{Q}} H(S,T,E,q) - H(S',T',E',q').$$

We bound the entropic radius of $P$ by the fact that $(P, S, T, E, q) \in \mathcal{Q}$ implies $\mathbf{1}^\top P \mathbf{1} = 1$, and thus

$$\max_{(P,S,T,E,q),(P',S',T',E',q')\in\mathcal{Q}} H(P) - H(P') \leq \log(n^2).$$

Likewise, the entropic radius of $(S, T, E, q)$ relies on the fact that $(P, S, T, E, q) \in \mathcal{Q}$ implies $\mathbf{1}^\top (S + T + E)\mathbf{1} + q \leq 3\varepsilon$, and thus

$$\max_{(P,S,T,E,q),(P',S',T',E',q')\in\mathcal{Q}} H(S,T,E,q) - H(S',T',E',q') \leq 3\varepsilon \log(3nd + 1),$$

and thus, one has

$$R_H \leq \log(n^2) + 3\varepsilon \log(3nd + 1).$$

Let $(P_\eta^\star, S_\eta^\star, T_\eta^\star, E_\eta^\star, q_\eta^\star)$ be the optimal solution to equation 6. For $\eta \geq \frac{1+3\varepsilon(1+\log(3nd+1))}{\Delta} > \frac{R_1+R_H}{\Delta}$, one has

$$\|P^\star - P_\eta^\star\|_1 \leq \|(P^\star, S^\star, T^\star, E^\star, q^\star) - (P_\eta^\star, S_\eta^\star, T_\eta^\star, E_\eta^\star, q_\eta^\star)\|_1$$

$$\leq 2R_1 \exp\left(-\eta\frac{\Delta}{R_1} + 1 + \frac{R_H}{R_1}\right)$$

$$= 2R_1 \exp\left(\frac{R_H - \eta\Delta}{R_1} + 1\right)$$

$$\leq 2(1 + 3\varepsilon) \exp\left(\frac{R_H - \eta\Delta}{1 + 3\varepsilon} + 1\right)$$

$$= 2(1 + 3\varepsilon) \exp\left(\frac{2\log(n) + 3\varepsilon\log(3nd+1) - \eta\Delta}{1 + 3\varepsilon} + 1\right)$$

$$\leq 6n^2(1 + 3\varepsilon) \exp\left(\frac{-\eta\Delta + 3\varepsilon\log(3nd+1)}{1 + 3\varepsilon}\right),$$

where the third inequality is because $R_H - \eta\Delta \leq 0$, and the last inequality holds because $\exp(\frac{2\log(n)}{1+3\varepsilon} + 1) \leq \exp(2\log(n) + 1) \leq 3n^2$. $\qquad\square$

## C   DERIVATION OF DUAL FORM FOR MOT

We now show that the dual form in equation 7 can be obtained from the primal-dual form by eliminating the dual variables. Let $H$ be the entropy term with $H(M) = M \cdot \log(M)$. Define

$$L(P, S, T, E, q, \mathbf{x}, \mathbf{y}, A, B, u) = \frac{1}{\eta} H(P, S, T, E, q) + C \cdot P - \mathbf{x} \cdot (P\mathbf{1} - \mathbf{r}) - \mathbf{y} \cdot (P^\top \mathbf{1} - \mathbf{c})$$

$$- A \cdot (PV - E - W + S) - B \cdot (PV + E - W - T)$$

$$- u(\mathbf{1}^\top E\mathbf{1} + q - \varepsilon).$$

Then, we show that for the $f$ in equation 7 is indeed the dual potential, as one has

$$f(\mathbf{x}, \mathbf{y}, A, B, u) = \min_{P,S,T,E,q} L(P, S, T, E, q, \mathbf{x}, \mathbf{y}, A, B, u). \tag{9}$$

By rearranging the terms, one has

$$\min_{P,S,T,E,q} L(P,S,T,E,q,\mathbf{x},\mathbf{y},A,B,u)$$

$$= \min_{P} \frac{1}{\eta} H(P) - P \cdot \left(\mathbf{x}\mathbf{1}^\top + \mathbf{1}\mathbf{y}^\top + (A+B)V^\top - C\right)$$

$$+ \min_{S} \frac{1}{\eta} H(S) - S \cdot A + \min_{T} \frac{1}{\eta} H(T) - T \cdot (-B)$$

$$+ \min_{E} \frac{1}{\eta} H(E) - E \cdot (-A + B + u\mathbf{1}\mathbf{1}^\top) + \min_{q} \frac{1}{\eta} H(q) - qu$$

$$+ \mathbf{x} \cdot \mathbf{r} + \mathbf{y} \cdot \mathbf{c} + (A+B) \cdot W + u\varepsilon.$$

For $M$ of arbitrary size, one has $\min_M \frac{1}{\eta} H(M) - M \cdot D = -\frac{1}{\eta} \sum_{ij} \exp\left(\eta d_{ij} - 1\right)$. Thus, the calculation gives

$$\min_{P,S,T,E,q} L(P,S,T,E,q,\mathbf{x},\mathbf{y},A,B,u)$$

$$= -\frac{1}{\eta} \sum_{ij} \exp\left(\eta(-c_{ij} + \sum_{k\in[d]}(a_{ik} + b_{ik})v_{jk} + x_i + y_j) - 1\right)$$

$$-\frac{1}{\eta}\left[\sum_{i\in[n],k\in[d]} \exp(\eta a_{ik} - 1) + \exp(-\eta b_{ik} - 1) + \exp(\eta(u - a_{ik} + b_{ik}) - 1)\right]$$

$$-\frac{1}{\eta}\exp(\eta u - 1) + \sum_i x_i r_i + \sum_j y_j c_j + \sum_{i\in[n],k\in[d]}(a_{ik} + b_{ik})w_{ik} + \varepsilon u,$$

which coincides with the formula for $f$. As $L$ is concave in $P,S,T,E,q$ and concave in $\mathbf{x},\mathbf{y},A,B,u$, we can invoke the Von-Neumann minimax theorem, and thus obtaining the optimal solution to entropic MOT problem is equivalent to maximization over $f$.

## D    SUPER-MARTINGALE CONDITION UNDER ENTROPIC REGULARIZATION

This section details the treatment of super-martingale optimal transport (SMOT). In this case, the feasibility of the constraint $PV \geq W$ is typically mild. Moreover, $W$ can always be sufficiently decreased to reach feasibility. Thus, we work on this problem by performing the entropic linear programming for the LP in equation 2.

**Variational formulation of SMOT**    In this case, we introduce a primal-dual form

$$L(P,S,\mathbf{x},\mathbf{y},A) = \frac{1}{\eta} H(P,S) + C \cdot P - \mathbf{x} \cdot (P\mathbf{1} - \mathbf{r}) - \mathbf{y} \cdot (P^\top\mathbf{1} - \mathbf{c}) - A \cdot (PV - W - S),$$

where each dual variable $a_{ik}$ corresponds to the $(i,k)$-th constraint in the matrix inequality $PV \geq W$. By the same calculation as that of Appendix C, one has

$$\min_{P,S} L(P,S,\mathbf{x},\mathbf{y},A)$$

$$= \min_{P} \frac{1}{\eta} H(P) - P \cdot \left(\mathbf{x}\mathbf{1}^\top + \mathbf{1}\mathbf{y}^\top + AV^\top - C\right)$$

$$+ \mathbf{x} \cdot \mathbf{r} + \mathbf{y} \cdot \mathbf{c} + A \cdot W + \min_{S} \frac{1}{\eta} H(S) + S \cdot A.$$

Thus, taking $g(\mathbf{x},\mathbf{y},A) = \min_{P,S} L(P,S,\mathbf{x},\mathbf{y},A)$, one then has the dual problem as follows:

$$\max g(\mathbf{x},\mathbf{y},A) = -\frac{1}{\eta} \sum_{ij} \exp\left(\eta(-C_{ij} + \sum_{k\in[d]}(a_{ik})v_{jk} + x_i + y_j) - 1\right)$$

$$+ \sum_i x_i r_i + \sum_j y_j c_j + \sum_{i\in[n],k\in[d]}(a_{ik})w_{ik} - \frac{1}{\eta} \sum_{i\in[n],k\in[d]} \exp(-\eta a_{ik} - 1). \tag{10}$$

**Sparsity and approximate sparsity of Hessian**  We discuss sparsity patterns of the Hessian matrix $\nabla^2 g$ for the algorithmic development of SMOT. Let $P = \exp\left(\eta(-C + \mathbf{x}\mathbf{1}^\top + \mathbf{1}\mathbf{y}^\top + AV^\top) - 1\right)$ be the intermediate transport plan corresponding to the current dual variable $(\mathbf{x}, \mathbf{y}, A)$. We remark that the blocks $\nabla_A^2 g, \nabla_\mathbf{x}\nabla_A g$ both have a block structure of diagonal matrices. Writing $A = [\mathbf{a}_1, \ldots, \mathbf{a}_k]$, one has

$$\nabla_\mathbf{x}\nabla_{\mathbf{a}_k} g = -\eta \operatorname{diag}(P\mathbf{v}_k),$$

and likewise $\nabla_{\mathbf{a}_k}\nabla_{\mathbf{a}_{k'}} g$ are diagonal matrices.

For the approximate sparsity, we note that $\nabla_\mathbf{y}\nabla_A g, \nabla_\mathbf{y}\nabla_\mathbf{x} g$ are dense matrices defined by $P$, as one has

$$\nabla_\mathbf{y}\nabla_\mathbf{x} g = -\eta P^\top, \quad \nabla_\mathbf{y}\nabla_{\mathbf{a}_k} g = -\eta \operatorname{diag}(\mathbf{v}_k)P^\top,$$

which shows that one can obtain sparse approximation of $\nabla^2 g$ by sparse approximation of $P$.

**Sinkhorn-type algorithm**  Due to the sparsity analysis, one can define $\mathbf{h} = (\mathbf{x}, A)$ and perform alternating maximization on $(\mathbf{y}, \mathbf{h})$. The Sinkhorn-type algorithm is summarized in Algorithm 4.

---

**Algorithm 4** Sinkhorn-type algorithm for entropic SMOT

---

**Require:** $g, \mathbf{x}_{\text{init}} \in \mathbb{R}^n, \mathbf{y}_{\text{init}} \in \mathbb{R}^n, A_{\text{init}}, N, i = 0, N_\mathbf{h} = 3$
1:  $\mathbf{y} \leftarrow \mathbf{y}_{\text{init}}, \mathbf{h} \leftarrow (\mathbf{x}_{\text{init}}, A_{\text{init}})$            ▷ Initialize dual variable
2:  **while** $i < N$ **do**
3:     $i_h \leftarrow 0, i \leftarrow i + 1$
4:     `# Column scaling step`
5:     $(\mathbf{x}, A) \leftarrow \mathbf{h}$
6:     $P = \exp\left(\eta(-C + AV^\top + \mathbf{x}\mathbf{1}^\top + \mathbf{1}\mathbf{y}^\top) - 1\right)$
7:     $\mathbf{y} \leftarrow \mathbf{y} + \left(\log(c) - \log(P^\top\mathbf{1})\right)/\eta$
8:     `# `$\mathbf{h}$` variable update step`
9:     **while** $i_h < N_\mathbf{h}$ **do**
10:       $\Delta\mathbf{h} = -\left(\nabla_\mathbf{h}^2 g\right)^{-1}\nabla_\mathbf{h} g$          ▷ Obtain search direction
11:       $\alpha \leftarrow \text{Line\_search}(g, \mathbf{h}, \Delta\mathbf{h})$
12:       $\mathbf{h} \leftarrow \mathbf{h} + \alpha\Delta\mathbf{h}$
13:       $i_h \leftarrow i_h + 1$
14:     **end while**
15: **end while**
16: Output dual variables $(\mathbf{x}, \mathbf{y}, A)$.

---

We remark that the algorithm is almost identical to Algorithm 1 except for slight modification. We take $N_\mathbf{h} = 3$ in this work. The per-iteration complexity is $O(n^2)$.

**Sinkhorn-Newton-Sparse**  Due to the analysis above, one sees that the approximate sparsity of $\nabla^2 g$ relies on the approximate sparsity of $P$, which in turn relies on the approximate sparsity of the entropically optimal SMOT solution $P_\eta^\star$. A slight modification of the analysis in Weed (2018) would likewise show that $P_\eta^\star$ is exponentially close to the optimal LP solution $P^\star$, which has at most $2n - 1 + nd$ nonzero entries assuming uniqueness of the LP in equation 2. By running sufficient iterations of the Sinkhorn-type algorithm, one has $P \approx P_\eta^\star$, and thus one can likewise introduce the Sinkhorn-Newton-Sparse (SNS) algorithm, whereby one runs the Sinkhorn-type algorithm for a few iterations, and one then switches to sparse Newton iteration. We summarize the algorithm in Algorithm 5. The per-iteration complexity is $O(n^2)$.

For completeness, we detail the Hessian approximation implementation. For $\text{Sparsify}(\nabla^2 g, \rho)$ in Algorithm 2, we keep the first $\lceil\rho n^2\rceil$ largest terms in the transport matrix $P$ and obtain the resulting sparsification $P_{\text{sparse}}$. Similar to the MOT case, by the discussion given, we only need to perform the sparsification procedure for the blocks $\nabla_\mathbf{y}\nabla_A g, \nabla_\mathbf{y}\nabla_\mathbf{x} g$ by replacing the role of $P$ with that of $P_{\text{sparse}}$.

**Algorithm 5** Sinkhorn-Newton-Sparse for entropic SMOT

**Require:** $g, \mathbf{x}_{\text{init}} \in \mathbb{R}^n, \mathbf{y}_{\text{init}} \in \mathbb{R}^n, A_{\text{init}}, N_1, N_2, \rho, i = 0$
1: # Warm initialization stage
2: Run Algorithm 4 for $N_1$ iterations to obtain warm initialization $(\mathbf{x}_{\text{init}}, \mathbf{y}_{\text{init}}, A_{\text{init}})$.
3: $z \leftarrow (\mathbf{x}_{\text{init}}, \mathbf{y}_{\text{init}}, A_{\text{init}})$                                   ▷ Initialize dual variable
4: # Newton stage
5: **while** $i < N_2$ **do**
6:     $H \leftarrow \text{Sparisfy}(\nabla^2 g, \rho)$                      ▷ Sparse approximation of $\nabla^2 g$.
7:     $\Delta \mathbf{z} \leftarrow -H^{-1}(\nabla g(\mathbf{z}))$                 ▷ Solve sparse linear system
8:     $\alpha \leftarrow \text{Line\_search}(g, \mathbf{z}, \Delta \mathbf{z})$              ▷ Line search for step size
9:     $\mathbf{z} \leftarrow \mathbf{z} + \alpha \Delta \mathbf{z}$
10:     $i \leftarrow i + 1$
11: **end while**
12: Output dual variables $(\mathbf{x}, \mathbf{y}, A) \leftarrow \mathbf{z}$.

