# OpenReview forum: "An efficient algorithm for entropic optimal transport under martingale-type constraints"
_ICLR.cc/2025/Conference — Submitted to ICLR 2025_

### Official Review · Reviewer_myZN · 2024-10-31

**Soundness:** 3
**Presentation:** 2
**Contribution:** 2
**Rating:** 5
**Confidence:** 4

**Summary:**

This work introduces novel computational methods for entropic optimal transport (OT) problems under martingale-type conditions. The OT problems under martingale-type conditions are discussed up to the middle of page 5. Entropic formulation starts in (6) on page 5. Theorem 1 gives an inequality that justifies the entropic regularization formulation. Then optimizing the dual problem is proposed in Section 4, which is consistent with existing approaches. Sinkhorn-type algorithm is proposed. Given the structure of the objective function in dual, the Sinkhorn-type algorithm is reasonable. The sparsity of Hessian is noticed. So, the resulting algorithm is fast.

**Strengths:**

The literature of numerically solving OT problem is well presented.

**Weaknesses:**

Entropic regularization leads to a dual with similar properties to other OT problems. The resulting algorithm is a standard algorithm in the OT literature, the Sinkhorn-type algorithm. It seems that this paper isn't very novel. It's too similar to the existing numerical considerations for OT problems. The martingale-type conditions did not require new mathematical analysis to come up with an algorithm.

An observation is that in the dual problem (7), authors should explain how these variables connect with the variables in the primal problem (6). This may help the presentation of the paper.

The title of the paper is "An Efficient Algorithm for...". However, the paper spends multiple pages on martingale-type conditions and related problems. The numerical consideration is mainly summarized from the middle of page 5 to the middle of page 6. It sounds like the main work is to figure out the dual objective function and realize it is a convex function. The algorithmic analysis is thin, and this seems to be an inconsistency by this reviewer.

**Questions:**

No.

---

> ### Author Response · Authors · 2024-11-14
> **Reply to the reviewer**
>
> The authors thank the reviewer for the reviewer's time. If the response addresses some of the reviewer's concerns, we kindly suggest that the reviewer consider increasing the score.
>
> ### Comment 1
> *Entropic regularization leads to a dual with similar properties to other OT problems. The resulting algorithm is a standard algorithm in the OT literature, the Sinkhorn-type algorithm. It seems that this paper isn't very novel. It's too similar to the existing numerical considerations for OT problems. The martingale-type conditions did not require new mathematical analysis to come up with an algorithm.*
>
> **Response:**
>
> The author would like to provide some clarifying remarks about the novelty. The first main novelty of this work is that the entropic regularization is based on an LP with approximate martingale condition. As the LP with exact martingale condition may not have any feasible solution, the approximate constraint satisfaction formulation in equation (4) is necessary. One of the main novelties of this work is to use the entropic LP formulation based on the LP in equation (4). While fairly natural, the authors think the proposed approach is the correct numerical treatment for entropic MOT. We can summarize this contribution by concluding that the authors identified the correct entropic formulation for MOT.
>
> Secondly, the APDAGD algorithm is quite strong and the design of an algorithm that outperforms APDAGD is a rather significant undertaking. The Sinkhorn-type algorithm is based on a novel analysis of the exact sparsity structure of the Hessian matrix, and the SNS algorithm is based on the approximate sparsity structure of the Hessian matrix, which are both novel observations of the newly proposed entropic formulation. The numerical performance is quite strong and the two algorithms' performance is much better than APDAGD. This design of a fast algorithm is a significant achievement for MOT and allows the computational MOT to catch up with the development of Sinkhorn in computational OT.
>
> ### Comment 2
> *An observation is that in the dual problem (7), authors should explain how these variables connect with the variables in the primal problem (6). This may help the presentation of the paper.*
>
> **Response:**
>
> The authors thank the reviewer for the suggestion. Following the suggestion, we added further remarks to the dual variables.
>
>
> ### Comment 3
>
> *The title of the paper is "An Efficient Algorithm for...". However, the paper spends multiple pages on martingale-type conditions and related problems. The numerical consideration is mainly summarized from the middle of page 5 to the middle of page 6. It sounds like the main work is to figure out the dual objective function and realize it is a convex function. The algorithmic analysis is thin, and this seems to be an inconsistency by this reviewer.*
>
> **Response:**
>
> The authors thank the reviewer for the remark and the authors agree that the numerical algorithm appears only at the second half of the main text. As mentioned in the reply to Comment 1, the entropic formulation for MOT is not established and a large portion of this manuscript goes into developing the best entropic formulation for MOT. Thus, figuring out what primal formulation to use is a quite important design choice for MOT.
> Moreover, the first half of the work also carries the important message that all row-wise constraints, such as certain kinds of fairness or diversity constraints, would be encoded by a martingale constraint. As the use of MOT in machine learning is a relatively nascent field, the writing also focuses on how fairness or diversity constraints manifest as MOT problems.
>
> In terms of the composition of the paper, however, we would like to remark that Appendix A contains the implementation detail for the APDAGD algorithm, and Appendix D contains the implementation detail for super-martingale optimal transport problems. If one takes those two sections into consideration, the focus on the numerical algorithm would be quite heavy. The authors do agree with the reviewer that the analysis of this work is rather light. The algorithmic analysis is outside the scope of this work, but we would like to comment that the  APDAGD algorithm has a robust convergence guarantee, and the Sinkhorn-type algorithm significantly outperforms it in practice. Thus, we do anticipate follow-up work to provide good numerical performance guarantee to our approach.

---

> > ### Comment · Reviewer_myZN · 2024-11-27
> > **Thank you for your responses!**
> >
> > Many thanks for your response! Given my perception of this work's contribution, I am inclined to keep my score unchanged.

---

### Official Review · Reviewer_8jb4 · 2024-11-01

**Soundness:** 3
**Presentation:** 3
**Contribution:** 2
**Rating:** 3
**Confidence:** 4

**Summary:**

In this article, the author(s) consider the martingale optimal transport (MOT) problem, and develop a Sinkhorn-type algorithm for solving a relaxed version of MOT. A sparse Newton method is also used to accelerate the Sinkhorn-type algorithm for better convergence speed.

**Strengths:**

The article is well organized, and the overall motivation and story is clear. The MOT problem considered is an extension to the well-known OT problem, and developing efficient algorithm for solving MOT is helpful.

**Weaknesses:**

1. One of my major concerns for this article is the necessity of using a Sinkhorn-type algorithm for solving the entropic MOT problem. The Sinkhorn algorithm is efficient partly because it has closed-form formulas for the two alternating minimization steps. However, in the entropic MOT problem, the author(s) show that the $g$ variables need to be updated using Newton's method. If this is the case, what is the benefit of using the alternating minimization method? We can simply use limited-memory quasi-Newton methods, e.g., L-BFGS, to solve problem (7): L-BFGS is well tested and practically useful for smooth unconstrained problems, the per-iteration cost is also $O(n^2)$, and it has the benefit of avoiding an inner loop for sub-problems.

2. In the abstract, the author(s) state that "As exact martingale conditions are typically infeasible, we adopt entropic regularization to find an approximate constraint satisfied solution". However, I do not think the logic here is correct. The entropic regularization is not related to the feasibility of the solution, but is used to smooth linear programming (LP) problems for faster computation. The infeasibility issue is addressed by the relaxation of constraints given in equation (4).

3. It is good to have Theorem 1 for the approximation error, but is it a direct application of Corollary 9 of [1]? Because [1] provides analysis for general LP problems, and once you have transformed the problem into a standard form, the results should automatically hold.

4. The sparse Newton algorithm is a good addition to the Sinkhorn-type algorithm, but I feel it largely overlaps with the previous work [2]. Is it simply an application of [2] to entropic MOT?

5. For the sparse Newton method, the author(s) mention the super-exponential convergence rate, but I do not find any theoretical guarantee. So is it only from empirical observation? Also, I doubt whether the sparsified $H$ is guaranteed to be invertible. If not, how do you address it?

[1] Weed, J. (2018). An explicit analysis of the entropic penalty in linear programming. Conference On Learning Theory.

[2] Tang, X., Shavlovsky, M., Rahmanian, H., Tardini, E., Thekumparampil, K. K., Xiao, T., & Ying, L. (2024). Accelerating Sinkhorn algorithm with sparse Newton iterations. International Conference on Learning Representations.

**Questions:**

See the "Weaknesses" section.

---

> ### Author Response · Authors · 2024-11-14
> **Reply to the reviewer (Part 1)**
>
> The authors thank the reviewer for the reviewer's time. If the response addresses some of the reviewer's concerns, we kindly suggest that the reviewer consider increasing the score.
> ### Comment 1
>
> *One of my major concerns for this article is the necessity of using a Sinkhorn-type algorithm for solving the entropic MOT problem. The Sinkhorn algorithm is efficient partly because it has closed-form formulas for the two alternating minimization steps. However, in the entropic MOT problem, the author(s) show that the $g$ variables need to be updated using Newton's method. If this is the case, what is the benefit of using the alternating minimization method? We can simply use limited-memory quasi-Newton methods, e.g., L-BFGS, to solve problem (7): L-BFGS is well tested and practically useful for smooth unconstrained problems, the per-iteration cost is also $O(n^2)$, and it has the benefit of avoiding an inner loop for sub-problems.*
>
> **Response:**
>
> The authors agree that the L-BFGS algorithm is worth considering. We have tried numerical experiments with L-BFGS, but the results were not quite desirable, and as a result we left the part out of the text for simplicity. To fully address the reviwer's question, we will add a section in the appendix showcasing the comparison between the Sinkhorn-type algorithm and L-BFGS before the end of the discussion session.
>
> **Update on Nov 15: We have an update on the status of L-BFGS in a follow-up reply titled "Summary of result on L-BFGS"**
>
> ### Comment 2
>
> *In the abstract, the author(s) state that "As exact martingale conditions are typically infeasible, we adopt entropic regularization to find an approximate constraint satisfied solution". However, I do not think the logic here is correct. The entropic regularization is not related to the feasibility of the solution, but is used to smooth linear programming (LP) problems for faster computation. The infeasibility issue is addressed by the relaxation of constraints given in equation (4).*
>
> **Response:**
>
> The authors would like to clarify on this point, this work is based on an LP formulation which asks the solution to approximately satisfy the martingale constraint. This is because an LP formulation with exact constraint satisfaction will typically be infeasible in the sense that the feasibility set is empty. This LP formulation is in equation (4) in the main text. The infeasibility refers to the infeasibility of the LP, which is highly non-trivial. This infeasibility of exact constraint satisfaction is a core feature in MOT, and in contrast this infeasibility doesn't occur in OT.
>
> Based on the LP formulation, we provide an associated entropically regularized problem in equation (6). The motivation here is that the LP in (4) is computationally hard to solve. In contrast, the entropic regularization leads to practical algorithms with $O(n^2)$ iteration complexity.
>
> ### Comment 3
> *It is good to have Theorem 1 for the approximation error, but is it a direct application of Corollary 9 of [1]? Because [1] provides analysis for general LP problems, and once you have transformed the problem into a standard form, the results should automatically hold.*
>
> **Response:**
>
> The authors agree with the reviwer that the proof is straightforward. The entropic regularization formulation for MOT is a novel setting proposed in this work, and the theorem is included for completeness.
>
>
> ### Comment 4
> *The sparse Newton algorithm is a good addition to the Sinkhorn-type algorithm, but I feel it largely overlaps with the previous work [2]. Is it simply an application of [2] to entropic MOT?*
>
>
> **Response:**
>
> The fact that sparse Newton iteration can be done in $O(n^2)$ complexity comes from a highly non-trivial observation of the sparsity pattern of the Hessian matrix. While [2] claims the SNS algorithm would have $O(n^2)$ complexity for $O(1)$ constraints, we show in this work that special constraints structure would permit an $O(n)$ number of constraints to be allowed and one could still have an algorithm with $O(n^2)$ iteration complexity.

---

> > ### Author Response · Authors · 2024-11-14
> > **Reply to the reviewer (Part 2)**
> >
> > ### Comment 5
> > *For the sparse Newton method, the author(s) mention the super-exponential convergence rate, but I do not find any theoretical guarantee. So is it only from empirical observation? Also, I doubt whether the sparsified $H$ is guaranteed to be invertible. If not, how do you address it?*
> >
> > **Response:**
> > The observation for super-exponential convergence is empirical. The sparsified $H$ follows a very specific truncation structure. Effectively, the sparsified Hessian is the Hessian matrix of a concave function plus a non-positive diagonal matrix. This ensures that the Hessian is negative semi-definite. So any slight regularization of the Hessian by the identity matrix will guarantee an invertible Hessian. We remark that the Hessian matrix regularization for convex optimization is well-established and falls under the name of the Levenberg–Marquardt (LM) algorithm. For our case, we use GMRES for the linear system solving, which provides sufficient numerical stability that the algorithm performs well typically, and we have not found the need to apply LM. If we apply regularization by a small identity matrix as in the LM algorithm, it could also potentially improve numerical performance.

---

> > > ### Author Response · Authors · 2024-11-15
> > > **Summary of result on L-BFGS**
> > >
> > > We update the reviewer on the status of running the L-BFGS algorithm. We found out that the L-BFGS algorithm would not work for the MOT dual formulation. We remark that it might still be possible for some variants of L-BFGS to work, but the authors think that the method would only work after quite substantial modification to the MOT problem formulation or to the L-BFGS algorithm's core logic.
> > >
> > > We use the random assignment experiment with the parameter setting as set in the manuscript. We use the same initialization doubling strategy for the warm initialization. As for the methods, we compare between the following three methods: (a) L-BFGS, (b) APDAGD, and (c) Sinkhorn-Newton-Sparse algorithm. The L-BFGS method terminate after two iterations and the optimizer declares that the function value change is too small, which forces the optimizer to terminate. The optimizer used is implemented by the scipy package, and so the only information we provide to the scipy package is the function call and the gradient call. Even by tuning the tolerance thresholds, we were not able to have the L-BFGS running.
> > >
> > > Subsequently, we changed to a simpler setting where we set setting the $\eta$ value to be $\eta = 100$, and we use the TV distance as the error metric. After 25 iterations, the Sinkhorn-Newton-Sparse algorithm is able to converge to the entropic optimal solution with error $ < 10^{-7}$. After 60 iterations, the APDAGD algorithm reaches an error of $ \approx 10^{-2}$. However, L-BFGS has an error of $ \approx 10^{-1}$.
> > >
> > > As such, we conclude that L-BFGS fails to improve the dual potential in hard instances, and it performs much worse than APDAGD and Sinkhorn-Newton-Sparse in easier instances.
> > >
> > > The authors have decided to not include the performance of L-BFGS in the revised manuscript. **The authors only show that L-BFGS has bad performance under the current experiment setting with the default implementation.** However, the L-BFGS method has a lot of design choices that are unexplored, and also the parameterization of the dual potential could be changed to make the optimization landscape better for L-BFGS. Thus, the authors decide to not include the result in the manuscript, as the our exploration on L-BFGS, while quite extensive, is not exhaustive.
> > >
> > > **If the response addresses some of the reviewer's concerns, we kindly suggest that the reviewer consider increasing the score.**

---

> ### Comment · Reviewer_8jb4 · 2024-11-23
>
> I thank the author(s) for their time and efforts to provide the responses, which clarify several questions I had. However, I still have the following concerns.
>
> 1. For the comparison with L-BFGS, since the author(s) do not release any code, I implemented a version available at https://colab.research.google.com/drive/1JwFN7g3pYmEtNe5c96MM30DJ0qAxF6CO?usp=sharing. This code reproduces the first experiment in the main article, and the result is as follows:
>
> For $\eta=10$,
>
> | Iter | L-BFGS       | BCD         |
> |------|--------------|-------------|
> | 0    | f = 2473.191 | f = 1.168865 |
> | 100  | f = 1.158457 | f = 1.157977 |
> | 200  | f = 1.157724 | f = 1.157831 |
> | 300  | f = 1.157245 | f = 1.157774 |
> | 400  | f = 1.157226 | f = 1.157728 |
> | 500  | f = 1.157225 | f = 1.157685 |
>
> For $\eta=100$,
>
> | Iter | L-BFGS        | BCD           |
> |------|---------------|---------------|
> | 0    | f = 9.6827432 | f = 0.0952288 |
> | 100  | f = 0.0929696 | f = 0.0926257 |
> | 200  | f = 0.0913372 | f = 0.0914078 |
> | 300  | f = 0.0910952 | f = 0.0910868 |
> | 400  | f = 0.0909456 | f = 0.0909776 |
> | 500  | f = 0.0909007 | f = 0.0909416 |
>
> I am not saying that L-BFGS will universally outperform block coordinate descent (BCD), but at least it should not perform as badly as the author(s) claim.
>
> 2. In the response to Comment 2, I think the author(s) exactly explained what I wanted to say. The logic I understand is that [MOT with equality constraint may be infeasible] => [Relax MOT with inequality constraint $\Vert PV-W \Vert_1\le\varepsilon$ ] => [Use entropic regularization to solve LP problem]. This means that dealing with infeasibility only happens in the first step, and is unrelated to entropic regularization. However, the original text is "As exact martingale conditions are typically infeasible, we adopt entropic regularization to find an approximate constraint satisfied solution", which seemingly suggests that the infeasibility issue is addressed by entropic regularization.
>
> 3. For Comment 4 and Comment 5, I think it is easy to show that the density of the Hessian matrix mostly depends on the transport plan $P$, and what I mean is that the truncation method for $P$ is the same as that in [2]. Also, I feel it is incorrect to say that "the sparsified Hessian is the Hessian matrix of a concave function plus a non-positive diagonal matrix". It is true that the real Hessian matrix is compute from a concave function, but the truncated version is not.

---

> ### Author Response · Authors · 2024-11-23
>
> The authors thank the reviewer for the effort in reproducing the code. authors would like to provide further clarification.
>
> For comment 1, the author would like to comment that the case of the option pricing example can indeed be run by L-BFGS. In the random assignment case, however, the scipy L-BFGS package failed to run for $\eta = 1200$. Also, the author would like to emphasize that the focus of this work is on accuracy compared to the entropically optimal solutions. As the key starting point of this work is the entropic formulation of MOT under approximate constriant satisfaction, the authors were open to quasi-Newton methods as the main optimization strategy. **However, L-BFGS can only run on a subset of the problems considered, and it is oftentimes inferior to even APDAGD, which is behind the reason why the authors omit L-BFGS as the optimization method.** The authors do not mean that L-BFGS doesn't work for all instances, and the authors indeed report result of L-BGFS for $\eta = 100$.
>
> Moreover, the comparison BCD method the reviewer provided is not the method that we use, as our method uses the Newton's method for the g variable update.
>
> To further clarify on Comment 5 on the invertibility of the Hessian, the procedure where one truncates the (i,j)-th entry of the transport matrix P is equivalent to taking out the $\frac{1}{\eta}\exp(\eta(-c_{ij} + x_{i} + y_{j}) - 1)$ term in the dual formulation. If one takes out all of the $\frac{1}{\eta}\exp(\eta(-c_{ij} + x_{i} + y_{j}) - 1)$ terms in the dual formulation corresponding to truncated entries, then the Hessain matrix one obtains is still negative semi definite. For the aforementioned hessian matrix, adding a non-negative diagonal matrix will yield the Hessian matrix that we consider in our case.

---

> > ### Comment · Reviewer_8jb4 · 2024-11-26
> >
> > I thank the authors for the additional comments. To be clear, since for now we only want to show the convergence speed of BCD, it doesn't matter which method is used to solve the $g$ update. I used L-BFGS just for simplicity.
> >
> > I feel it would be much easier for the discussion if the authors could attach some code that runs the experiments. I understand that the authors may want to protect the core algorithm before the work is published, but at least the simulated data can be provided, so that reviewers can try some baseline methods and better understand what is going on.
> >
> > The claim on the invertibility of Hessian still does not fully convince me. Note that the expression $\frac{1}{\eta}\exp(\eta(-c_{ij} + x_{i} + y_{j}) - 1)$ is for the entropic-regularized OT without martingale constraints, and for MOT the dual function is much more
> > complicated. If the proof is trivial, adding such a proof in the appendix would be very helpful.

---

### Official Review · Reviewer_GyGM · 2024-11-04

**Soundness:** 2
**Presentation:** 1
**Contribution:** 1
**Rating:** 3
**Confidence:** 4

**Summary:**

The paper proposes Sinkhorn-type algorithms with sparse Newton iterations to solve entropic optimal transport under martingale-type constraints. Some numerical experiments have been shown to validate the efficiency of the proposed algorithms.

**Strengths:**

Focusing on optimal transport with martingale conditions is interesting, given its applications, such as model-free optimal pricing, as highlighted by the authors. Additionally, it would be valuable to explore extensions of Sinkhorn-type and other OT solvers to address this novel class of optimal transport problems.

**Weaknesses:**

1. The paper lacks clarity in its structure. It is strongly recommended that the authors clearly state the main message at the beginning of each section and subsection. This would improve transitions and prevent readers from feeling confused.

2. Although the topic is interesting, the absence of a theoretical convergence analysis for the proposed algorithm raises concerns about its suitability for publication in high-tier machine learning conferences.

3. While the authors claim that the proposed algorithm performs well in practice, they have only benchmarked it against APDAGD. Given the availability of other computational methods for solving MOT, such as those proposed by Guo (2019), a more comprehensive comparison with existing literature in the numerical experiments is needed to robustly demonstrate the algorithm’s efficiency.

**Questions:**

1. How do the authors determine the switching criteria for transitioning to Newton's method, and what justifications support their choice?

2. To ensure a fair comparison, could the authors include the runtime of the warm initialization in the numerical experiments?

3. Could the authors clarify the technical challenges involved in extending the Sinkhorn algorithm

---

> ### Author Response · Authors · 2024-11-14
>
> The authors thank the reviewer for the reviewer's time. If the response addresses some of the reviewer's concerns, we kindly suggest that the reviewer consider increasing the score.
>
> ### Comment 1
>
>
> *The paper lacks clarity in its structure. It is strongly recommended that the authors clearly state the main message at the beginning of each section and subsection. This would improve transitions and prevent readers from feeling confused.
> Although the topic is interesting, the absence of a theoretical convergence analysis for the proposed algorithm raises concerns about its suitability for publication in high-tier machine learning conferences.*
>
>
> **Response:**
>
>
> We would greatly appreciate if the reviewer could provide further context on where the clarity can be improved. While the algorithmic convergence analysis is not included in this work, we would like to mention that the APDAGD algorithm has a robust convergence guarantee, and the Sinkhorn-type algorithm significantly outperforms it in practice, and so we do hope subsequent follow-up work can support the strong numerical findings rigorously.
>
>
> ### Comment 2
>
>
> *While the authors claim that the proposed algorithm performs well in practice, they have only benchmarked it against APDAGD. Given the availability of other computational methods for solving MOT, such as those proposed by Guo (2019), a more comprehensive comparison with existing literature in the numerical experiments is needed to robustly demonstrate the algorithm’s efficiency.*
>
>
> **Response:**
>
>
> The entropic regularization formulation for MOT is novel, and for this setting the authors do not have benchmark algorithms other than general-purpose algorithms such as APDAGD. We mention that the MOT method proposed by Guo (2019) calls for the use of a general LP solver with an interior point method. We mention that the interior point method has an $O(n^3)$ per iteration complexity and is quite difficult to scale to large problems. In the numerical experiment section, we take the setting of $n = 800$, and the LP solver proposed by Guo (2019) would take very long to solve even in that relatively small problem size setting. One of the central aim of this work is to put forward a method that enjoys an $O(n^2)$ iteration complexity, and so our algorithm would not be directly comparable with Guo (2019), as they are in very different settings.
>
>
> ### Question 1
> *How do the authors determine the switching criteria for transitioning to Newton's method, and what justifications support their choice?*
>
>
>
> **Response:**
>
> The switching criteria is currently done by fixing a pre-selected integer for simplicity. One could use an adaptive method by checking for the magnitude of the gradient for the dual potential, and switching to the Newton iteration after the gradient is small. Having a small gradient will mean that the current dual variables will be close to the region where Newton's algorithm would enjoy super-exponential convergence. Typically, an earlier switch to the sparse Newton iteration would not be too disadvantageous, as any variational method will increase the dual potential and make the dual variables closer to the optimal point.
>
>
> ### Question 2
> *To ensure a fair comparison, could the authors include the runtime of the warm initialization in the numerical experiments?*
>
> **Response:**
>
> The runtime of the warm initialization is very mild. The warm initialization for the option pricing example and the assignment example both take about 4.5 seconds.
>
>
> ### Question 3
> *Could the authors clarify the technical challenges involved in extending the Sinkhorn algorithm*
>
> **Response:**
>
> The challenge is that the Sinkhorn's algorithm is done via matrix multiplication and matrix scaling, which are quite conventional numerical linear algebra routines. In contrast, this new setting calls for quite complicated linear system solving to perform the $g$ update step. Even though the Sinkhorn algorithm and the Sinkhorn-type algorithm both perform block-wise ascent methods, the Sinkhorn-type algorithm is more involved and has a longer runtime.

---

### Official Review · Reviewer_BnPL · 2024-11-04

**Soundness:** 3
**Presentation:** 3
**Contribution:** 3
**Rating:** 6
**Confidence:** 3

**Summary:**

This paper considers a linear program for the problem of optimal transport with martingale type constraints, which is in the form of standard Kantorocich relaxation with multiple additional linear constraints. Employing an entropic regularization, this work arrives at a dual (variational) framework, which is then solved by block coordinate ascend. A part of this procedure recovers the well-known Sinkhorn algorithm. However, an additional block appears due to the additional constraints, which is then maximized by the Newton's method. It is further shown that the underlying Hessian matrix is sparse, leading to less computations in the Newton steps.

**Strengths:**

The paper provides and discusses multiple examples that well justifies the underlying problem in various fields. In support of the choice of entropic regularization, the paper proves that the effect of this term vanishes at an exponential speed w.r.t the growth of the regularization parameter. The resulting algorithm is compared to a state-of-the-art first-orde method, which shows remarkable improvement in the speed of convergence.

**Weaknesses:**

I have few concerns related to the algorithmic choices and the theory that will explain tn the next part (questions).

The theoretical result is a streightforward generalization of the result in (Weed 2018), but still interesting.

The exaperiments certainly support that the algorithm is applicable to relatively large problems, but the setup of the experiments is still considered small in certain fields such as machine learning. However, this is a general limitation of the Kantorovich formulation, and is not necessarily a limitation for this paper.

**Questions:**

1- My main concern is that I am not sure if the choice of an entropy regularization makes sense for the additional constraints. For the standard OT problem, this choice is justified as the dual problem can be solved by exact block coordinate ascent, but if one needs to employ the Newton's method, then, why should not one use e.g. a logarithmic barrier (instead of entropic regularization), which also has the self-concordance property?

2- I appreciate the provided theoretical results, but I think that it still does not show how much the complexity grows with dimensions. A general concern about Theorem 1 is that although it is formulated as an exponential decay, it really shows the requirement that the regularization parameter grows proportionally with the inverse of the duality gap. In practice, the gap can be extremely small for large problems, leading to extremely large regularization parameters. As a result, it is interesting to see how the speed of convergence scales with the regularization parameter (is it linear, for example, as in (Altschuler 2017)?). The theoretical converghence analysis of the algorithm seems to be lacking.

---

> ### Author Response · Authors · 2024-11-14
> **Reply to the reviewer**
>
> The authors thank the reviewer for the reviewer's time. If the response addresses some of the reviewer's concerns, we kindly suggest that the reviewer consider increasing the score.
>
> ### Comment 1
>
>
> *My main concern is that I am not sure if the choice of an entropy regularization makes sense for the additional constraints. For the standard OT problem, this choice is justified as the dual problem can be solved by exact block coordinate ascent, but if one needs to employ the Newton's method, then, why should not one use e.g. a logarithmic barrier (instead of entropic regularization), which also has the self-concordance property?*
>
>
>
> **Response:**
>
> The authors agree with the reviewer that a log barrier can be used. Following this choice of log barrier, the associated Newton's method would not have a sparse approximation, and the Newton step would be of cost $O(n^3)$ per iteration. In contrast, the proposed Sinkhorn-type algorithm has an $O(n^2)$ iteration complexity. Thus, it is quite likely that using the entropic barrier for the coupling while using the log barrier for the constraints might be less desirable than simply using the interior point method for the LP in equation (4).
>
>
> ### Comment 2
>
> *I appreciate the provided theoretical results, but I think that it still does not show how much the complexity grows with dimensions. A general concern about Theorem 1 is that although it is formulated as an exponential decay, it really shows the requirement that the regularization parameter grows proportionally with the inverse of the duality gap. In practice, the gap can be extremely small for large problems, leading to extremely large regularization parameters. As a result, it is interesting to see how the speed of convergence scales with the regularization parameter (is it linear, for example, as in (Altschuler 2017)?). The theoretical converghence analysis of the algorithm seems to be lacking.*
>
> **Response:**
>
> The authors agree with the reviewer and would like to remark the exponential convergence analysis depends on the optimality gap, and thus could be less effective for particular cases. There is also a slower rate of $O(1/\eta)$ convergence, which would also be a good convergence result with a moderate regularization parameter. The authors do not have a concrete answer on the convergence guarantee for the proposed approach, but the numerical experiments we conduct indeed showcase good performance. We hypothesize that a thorough analysis would lead to a comparable guarantee as that of (Altschuler 2017). The manuscript also acknowledges this limitation in the conclusion section.

---

> > ### Comment · Reviewer_BnPL · 2024-11-27
> >
> > Many thanks for your response. I find the justification of the entropic regularization by the sparsity of Hessian interesting, but given the contribution of this work, I am inclined to keep my score unchanged.

---

### Meta-Review · Area_Chair_JTrk · 2024-12-17

**Metareview:**

The paper addresses entropic optimal transport (OT) under martingale-type constraints, proposing a Sinkhorn-type algorithm with sparse Newton iterations for computational efficiency. Despite its potential, the reviewers exposed several weaknesses. Concerns were raised about the lack of clarity in the paper’s structure, insufficient theoretical convergence analysis, and incomplete experimental comparisons, particularly with methods from prior works like Guo (2019) and L-BFGS variants. Additionally, doubts were expressed about the necessity of the Sinkhorn formulation, given the need for Newton iterations, as well as the accuracy of claims regarding Hessian sparsity and invertibility. While the paper presents a novel problem setting and offers practical improvements, the contributions appear incremental, and the empirical results are limited in scope. It cannot be accepted in the conference.

**Additional Comments On Reviewer Discussion:**

During the reviewer discussion, the main issues revolved around the lack of theoretical guarantees, unclear algorithmic justifications, and limited comparisons to relevant baselines. Without a rebuttal, these concerns remained unresolved, ultimately leading to the decision to reject the paper based on its current form and the consensus among reviewers regarding its shortcomings.

---

### Decision · Program_Chairs · 2025-01-22

Reject